# Identification of Anoikis-Related Subgroups and Prognosis Model in Liver Hepatocellular Carcinoma

**DOI:** 10.3390/ijms24032862

**Published:** 2023-02-02

**Authors:** Yutong Chen, Weiran Huang, Jian Ouyang, Jingxiang Wang, Zhengwei Xie

**Affiliations:** 1International Cancer Institute, Peking University Health Science Center, Beijing 100191, China; 2Department of Molecular and Cellular Pharmacology, School of Pharmaceutical Sciences, Peking University Health Science Center, Beijing 100191, China

**Keywords:** anoikis, TCGA, ICGC, prognostic model, immune infiltration

## Abstract

Resistance to anoikis is a key characteristic of many cancer cells, promoting cell survival. However, the mechanism of anoikis in hepatocellular carcinoma (HCC) remains unknown. In this study, we applied differentially expressed overlapping anoikis-related genes to classify The Cancer Genome Atlas (TCGA) samples using an unsupervised cluster algorithm. Then, we employed weighted gene coexpression network analysis (WGCNA) to identify highly correlated genes and constructed a prognostic risk model based on univariate Cox proportional hazards regression. This model was validated using external datasets from the International Cancer Genome Consortium (ICGC) and Gene Expression Omnibus (GEO). Finally, we used a CIBERSORT algorithm to investigate the correlation between risk score and immune infiltration. Our results showed that the TCGA cohorts could be divided into two subgroups, with subgroup A having a lower survival probability. Five genes (*BAK1*, *SPP1*, *BSG*, *PBK* and DAP3) were identified as anoikis-related prognostic genes. Moreover, the prognostic risk model effectively predicted overall survival, which was validated using ICGC and GEO datasets. In addition, there was a strong correlation between infiltrating immune cells and prognostic genes and risk score. In conclusion, we identified anoikis-related subgroups and prognostic genes in HCC, which could be significant for understanding the molecular mechanisms and treatment of HCC.

## 1. Introduction

Hepatocellular carcinoma (HCC), the most common form of liver cancer [1], is the fourth leading cause of mortality and the second most lethal malignant tumor worldwide [2]. Despite the fact that the availability of diagnostic tools is limited, such as histology and radiologic tests, there is still a great demand for swift, accurate and convenient methods for diagnosing HCC [3]. It is therefore critical to develop novel diagnostic methods to monitor the progression of cancer. Although clinically correlative biomarkers such as α-fetoprotein (AFP) are commonly used for HCC diagnosis, there is much controversy and limitations associated with their application [4]. In addition, copy number alteration of *FGF19*, *CCDN1*, *CDKN2A* and *CDKN2B* genes were found in 286 HCC patients, suggesting that these molecules are specific to the development of HCC [5]. Overall, it is of great significance to develop novel genes for the diagnosis and prognosis of HCC.

Anoikis is a form of programmed cell death that is triggered by the loss of interaction between the cell and extracellular matrix (ECM) [6,7]. In normal cells, this interaction is disrupted by anoikis-initiating molecules on the cell surface and glycosylated ECM proteins, leading to apoptosis and cell death. In contrast, tumor cells are protected by a “barrier” that prevents the activation of anoikis-initiating molecules and protects them from cell death, resulting in resistance to anoikis and promoting cell survival [8,9]. Recent research has uncovered molecular pathways and mechanisms that regulate anoikis resistance, including cell adhesion molecules, growth factors and signaling pathways inducing epithelial–mesenchymal transitions [6]. Downstream molecules in these pathways, such as the focal adhesion kinase [10], Src kinase [11], mitogen-activated protein kinase (MAPK) [12], ERK1/2 [13], Bcl-2 family [14], PI3K/Akt [15,16] and insulin-like growth factor receptors [17] have been shown to play an essential role in anti-apoptosis and pro-survival. In addition, specific molecules have been found to be involved in various types of cancer. For example, toll-like receptor (TLR), that recognizes damage-associated molecular patterns released from tumorigenic cells, could activate T cells and remove tumorigenic cells [18]. It was reported that the toll-like receptor 4 ligand, manganese superoxide dismutase, collagen XIII, nuclear factor κB (IκB) Kinase-ε (IKKε) and deleted in breast cancer-1 (DBC1) were associated with anoikis resistance in triple-negative breast cancer [19]. Trk kinase, the Hippo pathway and eEF-2 kinase were associated with glioma [20]. However, the molecular mechanisms and pathways of anoikis in the progression and invasion of HCC are still unknown. In the meantime, pharmacological compounds and molecular inhibitors have been used to treat metastatic tumors resulting from anoikis resistance. DZ-50, a novel quinazoline-based compound, induced anoikis to decrease the cell adhesion between the tumor and ECM and prevented tumor growth and neovascularization [21]. Glo1 inhibitors exerted an antitumor role by accumulating intracellular methylglyoxal to promote anoikis [22]. It is therefore crucial to uncover the cellular features and molecular mechanisms of anoikis in HCC, which could benefit the development of therapy and prognosis of HCC.

In this study, we applied comprehensive and multiscale bioinformatics analysis to identify the key regulators and essential prognostic genes in HCC. We applied an unsupervised cluster algorithm to cluster The Cancer Genome Atlas (TCGA) dataset based on differentially expressed overlapping anoikis-related genes. Then, we used weighted gene coexpression network analysis (WGCNA) to identify the most highly correlated genes and performed enrichment analysis in the Gene Ontology (GO) and Kyoto Encyclopedia of Genes and Genomes (KEGG) pathways to find potential biological pathways. We also utilized univariate Cox regression, Least absolute shrinkage and selection operator (LASSO) regression and multivariate Cox regression analyses to screen prognostic genes and constructed a prognostic risk model. Furthermore, we explored the correlation of infiltrating immune cells with prognostic genes and a risk score. In summary, our work identified novel regulator genes related to the tumor microenvironment of HCC, which could be significant in the development of prognostic genes for the diagnosis and treatment of HCC.

## 2. Results

### 2.1. Identification of Differentially Expressed Overlapping Anoikis-Related Genes

To identify differentially expressed overlapping anoikis-related genes, we first identified the differentially expressed genes (DEGs) in TCGA liver hepatocellular carcinoma (LIHC). The results showed that 6311 genes were differentially expressed (Figure 1A). Then, we found 448 anoikis-related genes from GeneCards database and finally screened 168 differentially expressed overlapping anoikis-related genes for further analysis (Figure 1B).

### 2.2. Identification of Anoikis-Related Subgroups

To identify anoikis-related subgroups, we performed unsupervised clustering using a consensus clustering algorithm to explore the possible clusters within TCGA-LIHC datasets, based on the expression of overlapping anoikis-related genes. The results showed that k = 2 was the best parameter for dividing LIHC datasets into the subgroups A and B (Figure 2A). Principal Component Analysis (PCA) indicated a clear classification between subgroup A and subgroup B (Figure 2B). The results were also verified using t-distributed stochastic neighbor embedding (t-SNE) and uniform manifold approximation and projection (UMAP) analysis (Appendix A). Furthermore, subgroup A had a lower survival probability than subgroup B (Figure 2C). The heatmap of expression of overlapping anoikis-related genes showed a difference between the two subgroups (Figure 2D). These results indicated that TCGA-LIHC can be classified based on anoikis-related genes.

### 2.3. Different Characteristics of Biological Behavior between Two Subgroups

To explore the different biological processes between the two subgroups, we performed Gene Set Variation Analysis (GSVA) to identify the different KEGG pathways and the Reactome pathway within the two subgroups. The results showed that subgroup A was highly enriched in Golgi cisternae pericentriolar stack reorganization, activation of NIMA kinases NEK9, NEK6 and NEK7 and condensation of prophase chromosomes. Subgroup A was also enriched in the dorsoventral axis formation, inositol phosphate metabolism and notch signaling pathway (Figure 2E). Moreover, we applied the ssGSEA (single-sample gene set enrichment analysis) algorithm to explore the level of immune infiltration between the two subgroups. The results indicated that subgroup A had a distinct pattern of immune infiltration compared to subgroup B (Figure 2F). Furthermore, subgroup A had a significantly higher abundance of immune cells—including activated CD4, IDC, MDSC, memory B cells, NK cells and TGD—than subgroup B, while subgroup B had a significant abundance of immune cells—including activated CD8, macrophages, mast cells, monocytes and PDC—than subgroup A (Figure 2G). These results showed that the two subgroups had different characteristics in the KEGG pathway, Reactome pathway and immune infiltration levels.

### 2.4. Identification of Highly Correlated Gene Module in TCGA-LIHC

To explore the highly correlated genes among the overlapping anoikis-related genes, we performed WCGNA to identify the highly correlated gene modules. First, we identified two samples as outliers based on a sample clustering tree and excluded them (Appendix A). We then set the soft thresholding power to 5 (Figure 3A) and identified three gene modules based on the gene dendrogram: the turquoise module, brown module and blue module (Figure 3C). A heatmap of the correlation between the different modules is also shown (Figure 3B). Among these gene modules, there was a strong correlation between tumor occurrence and the turquoise module (the coefficient was 0.62 and *p*-value was 8 × 10^−47^, Figure 3D). In addition, the gene significance (GS, i.e., the correlation between the genes and clinical traits) and module membership (MM, i.e., the correlation between the genes and modules) were highly correlated in the turquoise module, indicating that the genes in this module were most significantly associated with tumors (Figure 3E). Genes in the brown module were also significantly associated with tumors, while genes in the blue module were not associated with tumors (Appendix A). Finally, we extracted 75 genes from the turquoise module for further analysis.

To explore the role of genes in the turquoise module, we performed GO and KEGG pathway enrichment analyses. The results showed that these genes were highly enriched in GO–Biological Process (BP) terms related to the apoptotic process, as well as to the positive and negative regulation of the apoptotic process. Additionally, these genes were also highly enriched in positive regulation anoikis GO–BP terms (Figure 3F). In terms of GO–Cell Component (CC) terms, these genes were highly enriched in nucleus, cytosol and cytoplasm, and in GO–Molecular Function (MF) terms, they were enriched in protein binding, identical protein binding and protein kinase binding (Appendix A). KEGG pathway enrichment analysis showed that these genes were enriched in human immunodeficiency virus 1 infection, hepatitis B and hepatitis C, which was consistent with the development and progression of liver cancer (Figure 3G). Overall, the genes in the turquoise module were highly correlated with the occurrence, metastasis and development of liver hepatocellular carcinoma.

### 2.5. Identification of Anoikis-Related Gene Clusters

Next, we identified anoikis-related gene clusters based on the highly correlated genes in the turquoise module in the WGCNA analysis. Similarly to the anoikis-related subgroups, the LIHC dataset was divided into 3 clusters (Figure 4A), which was verified using PCA analysis (Figure 4B). Moreover, cluster A had a lower survival probability than clusters B and C (Figure 4C). A heatmap of gene expression shows the differences between the three clusters (Figure 4D).

### 2.6. Construction of Anoikis-Related Prognostic Risk Model

To construct an anoikis-related prognostic risk model, we randomly sampled 183 samples for the training dataset and 182 samples for the test dataset from the LIHC dataset. Using univariate Cox proportional hazard regression analysis, we applied a total of 75 genes to screen for candidate prognostic genes and identified 55 genes based on a *p*-value < 0.05. The top 20 genes are shown in Figure 5C. We then applied these genes to perform LASSO regression and identified five genes with the minimum lambda value of 0.1153418 (Figure 5A,B). The risk score was calculated as follows: Risk score = (expression of *BAK1** 0.041298989) + (expression of *BSG** 0.023046274) + (expression of *SPP1** 0.018401788) + (expression of *DAP3** 0.004892867) + (expression of *PBK** 0.067923316). Patients were divided into two groups based on the median value of the risk score. In the test dataset, patients with a high risk score tended to have a lower survival probability and die earlier than those with a low risk score (Figure 5D). In addition, the gene expression profiles of the prognostic genes were significantly different between the two risk groups (Figure 5E). Furthermore, the overall survival, survival time and gene expression profiles of the prognostic genes in the test dataset were consistent with those in the training dataset (Figure 5F,G).

Additionally, we predicted the overall survival in the training and test datasets. The areas under the time-dependent ROC curves (AUCs) at 1, 3 and 5 years were 0.81, 0.763 and 0.696 in the test dataset (Figure 6A) and 0.796, 0.727 and 0.826 in the training dataset, respectively (Appendix A). The expression of prognostic genes was significantly upregulated in patients with high risk compared to those with low risk in both the training and test datasets (Figure 6B and Appendix A). All prognostic genes were significantly associated with poor survival probability in the training dataset (Appendix A), but only *SPP1* and *BSG* were significantly associated with poor survival probability in the test dataset (Figure 6D). The nomogram showed that risk score played a critical role in predicting overall survival at 1, 2, 3, 4 and 5 years (Figure 6C). Overall, the anoikis-related prognostic risk model was established successfully and had a strong performance in predicting overall survival in liver cancer.

### 2.7. Validation of Anoikis-Related Prognostic Risk Model

ICGC and GEO datasets were applied to validate the performance of the anoikis-related prognostic risk model. The results showed that patients with a high risk score had lower survival probability and were more likely to die than those with a low risk score. Additionally, the prognostic genes in the high-risk group were significantly upregulated compared to those in the low-risk group, with the exception of *DAP3* (Figure 7A–D). Moreover, the AUCs for the prognostic model were 0.777, 0.733 and 0.853 in the ICGC dataset (Figure 7E), and *BAK1*, *BSG* and *PBK* were significantly correlated with poor survival probability (Appendix A). The different pattern of prognostic gene expression was also consistent with the TCGA dataset (Appendix A). These results were similar to those in the TCGA dataset, indicating that the anoikis-related prognostic risk model had a strong performance for predicting overall survival.

Additionally, the relative expression of prognostic genes was significantly upregulated in the tumor samples compared to the normal samples in GSE84402, with the exception of *SPP1* (Figure 7F). The same was true for GSE101685, with the exception of *BAK1* and *BSG* (Figure 7G). The protein levels of prognostic genes were also verified using the Human Protein Atlas (HPA) database, which showed that the protein expression of all prognostic genes was upregulated in the tumor samples compared to the normal samples (Appendix A).

### 2.8. Different Characteristics of Immune Cell Infiltration between Two Risk Groups

According to the results above, we found the distribution and relationship of two subgroups, three gene clusters, two risk groups and two clinical outcomes (Figure 8A). Moreover, we found a significant difference between two subgroups and three gene clusters in risk score (Figure 8B,C). Next, we performed the CIBERSORT algorithm to explore the correlation between infiltrating immune cells and risk group. The results showed that risk score had a significantly positive correlation with the monocyte (cor = 0.26, *p* ≤ 0.00001), macrophage M0 (cor = 0.15, *p* = 0.00363), M1 (cor = 0.22, *p* = 0.00002), M2 (cor = 0.15, *p* = 0.00527), resting NK cells (cor = 0.22, *p* = 0.00003), resting dendritic cells (cor = 0.21, *p* = 0.00006), activated mast cells (cor = 0.19, *p* = 0.00021), gamma delta T cells (cor = 0.18, *p* = 0.00064), plasma cells (cor = 0.17, *p* = 0.00135) and follicular helper T cells (cor = 0.13, *p* = 0.01375), and it had a significantly negative correlation with the eosinophils (cor = −0.22, *p* = 0.00003) and resting mast cells (cor = −0.16, *p* = 0.00212) (Figure 8D–G and Appendix A). The prognostic genes were also found to be highly correlated with most of the immune cells, with the exception of *DAP3* (Figure 8H).

## 3. Discussion

In this study, we identified the role of anoikis-related genes in the development and progression of HCC and constructed a prognostic risk model. We also explored the correlation between infiltrating immune cells and prognostic genes and risk score. TCGA samples were divided into two subgroups, with subgroup A having a lower survival probability. We used GSVA and ssGSEA to analyze the biological processes and infiltrating immune cells in these subgroups and found that subgroup A was enriched in Golgi cisternae pericentriolar stack reorganization, NIMA kinase activation and prophase chromosome condensation. We also found that subgroup A was enriched in dorsoventral axis formation, inositol phosphate metabolism and the notch signaling pathway. Using WGCNA, we screened for genes highly correlated with HCC and constructed a prognostic risk model. Finally, we analyzed the correlation between infiltrating immune cells and prognostic genes and the risk score. We found that the risk score had a significantly positive or negative correlation with most immune cells and the prognostic genes were highly correlated with most immune cells, with the exception of *DAP3*. In conclusion, our findings suggest that anoikis-related genes can be used to assess the prognostic significance and potential for immunotherapy of HCC.

The results of GSVA showed that subgroup A was enriched in the Reactome pathway related to Golgi cisternae pericentriolar stack reorganization, activation of the NIMA kinases NEK9, NEK6 and NEK7 and condensation of prophase chromosomes, which are all essential for mitosis. The NEK family of serine/threonine kinases was also identified as a potential biomarker for cancer [23]. Additionally, subgroup A was enriched in the anchoring of the basal body to the plasma membrane and release of apoptotic factors from the mitochondria, which were associated with anoikis. Subgroup A was also enriched in the KEGG pathway related to dorsoventral axis formation, inositol phosphate metabolism and the notch signaling pathway. Inositol phosphate had been linked to apoptosis and cancer [24], and the notch signaling pathway was known to regulate tumor development and the microenvironment [25]. However, there was little research on the relationship between dorsoventral axis formation and liver cancer, suggesting that anoikis-related genes may play a role in tissue and organ development, in addition to their involvement in HCC.

Immune cells can alter the liver environment and trigger chronic inflammation, eventually leading to the development of hepatocellular carcinoma [26]. In our study, the risk score had a significantly positive correlation with monocytes, macrophages, dendritic cells, natural killer cells (NK cells), T cells and plasma cells and had a negative correlation with eosinophils and mast cells, which was consistent with previous studies. It has been reported that immune cells such as CD4+T cells, CD8+T cells, NK cells, NKT cells, monocytes and macrophages were activated and participated in mediating liver inflammation during chronic HBV infection, which eventually promoted the development of HCC [27]. Additionally, studies have shown that an elevated proportion of myeloid-derived suppressor cells (MDSCs) in the liver promoted hepatocarcinogenesis in mice models [28]. Dendritic cells (DCs) have also been found in high numbers in the peripheral blood and liver of HCC patients [29]. The stimulation of T cells can reduce the secretion of IL-12 by DCs [30], and high levels of chemokines and mast cells have been found in tumor regions [31]. The two subgroups in our study showed different patterns of immune cell abundance, which may be due to differences in the tumor microenvironment. The development of HCC was delayed by the depletion of CD8+ T cells in mice models, and these mice had an increased incidence of HCC [32]. Additionally, the role of CD4+T cells and B lymphocytes in the development of HCC remains controversial [33].

Astrocyte elevated gene-1 (*AEG-1*) has been identified as an important oncogene that promotes anoikis resistance in HCC cells. AEG-1 promotes anoikis resistance by activating the PI3K/Akt pathway and upregulating the apoptosis protein BCL-2 and the phosphorylation of *Bad* [34]. In our study, we found that five novel prognostic genes (*BAK1*, *SPP1*, *BSG*, *PBK* and *DAP3*) were risk factors for anoikis resistance in HCC. *BAK1* is a multi-domain proapoptotic effector belonging to the *Bcl-2* family, which controls cell homeostasis via apoptosis [35]. The activation of *BAK1* permeabilizes the mitochondrial outer membrane to regulate apoptosis [36]. This suggests that anoikis may be related to mitochondrial outer membrane permeabilization. Interestingly, *BAK* expression was decreased in colorectal tumors and remained unchanged in most conditions [37], but increased *BAK* expression was strongly associated with a poor prognosis in patients with non-small cell lung cancer [38], which indicates that the role of *BAK* is unclear in different types of cancers. Consistent with our results, *BAK1* has been identified as a prognostic gene in pyroptosis-related HCC and is associated with persistent hepatitis B virus infection-related HCC [39,40]. In this study, we found that the expression of *BAK1* was upregulated in the high-risk group compared to the low-risk group. Furthermore, the results of the validation cohorts showed that the expression of *BAK1* was upregulated in the tumor samples compared to the normal samples.

Death-associated protein 3 (*DAP3*) is a classical anoikis marker and is necessary for the induction of anoikis [41]. It has been reported that interferon-beta promoter stimulator 1 (*IPS-1*) binds to *DAP3* and induces anoikis by activating caspase-8 [42]. *DAP3* was overexpressed in invasive glioblastoma cells [43]. However, the role of *DAP3* in the development and progression of other tumors is not well understood. Other prognostic genes in our study have been found to be involved in several types of cancers. For example, the expression of *BSG* has been found to change significantly in many tumors, including cholangiocarcinoma, colon adenocarcinoma and rectum adenocarcinoma. It has also been associated with the prognosis of eight cancers, including invasive breast carcinoma [44]. Intriguingly, *BSG* has been identified as a potential target of non-coding RNA in hepatocellular carcinoma tumorigenesis [45,46]. *SPP1* is involved in the tumor microenvironment of pancreatic and prostate cancer [47,48]. It has also been identified as a novel biomarker in nonalcoholic steatohepatitis-related HCC [49]. The expression of *PBK* has been associated with the prognoses of lung cancer [50], colorectal cancer [51] and gastric cancer [52]. In the progression of HCC, the overexpression of *PBK* has been found to promote HCC cell migration and invasion by activating the ETV4/urokinase-type plasminogen activator receptor signaling pathway [53]. Yang et al. showed that *PBK* enhanced the metastasis of HCC by activating β-catenin signaling [54]. However, further research is needed to fully understand the molecular mechanisms involved.

Although our study identifiy anoikis-related subgroups and prognostic genes in HCC for the first time, there are several limitations. First, the overall number of cohorts and sequencing data is limited. Second, the clinicopathologic characteristics of patients are limited, so more practical and valuable factors are needed to predict the survival rate at 1–5 years. Finally, the study lacks basic experiments to prove the expression of prognostic genes in liver cancer cell lines, which needs further investigation.

In summary, this study identified anoikis-related subgroups and constructed a prognostic risk model of HCC. The results showed that five prognostic genes were identified and were highly correlated with the occurrence of tumor and immune cell infiltration. The prognostic risk model had a strong and effective performance in predicting the overall survival of HCC. Overall, our findings have great significance for investigating the molecular pathways and mechanisms involved in HCC and for developing treatments and prognoses of HCC.

## 4. Materials and Methods

### 4.1. Data Collection

The gene expression profiles for TCGA liver hepatocellular carcinoma (LIHC), including 50 normal and 371 tumor samples, were downloaded using the R package “TCGAbiolinks”. The phenotype and survival data of the LIHC cohorts were downloaded from UCSC Xena (https://xenabrowser.net/datapages/, (accessed on 21 September 2022)). The DEGs were analyzed using the R package “TCGAbiolinks”. The *p*-values were adjusted based on the false discovery rate (FDR) correction method, and the DEG cutoff was set as |log_2_FC| > 1 and adjusted to a *p*-value < 0.05.

The gene expression profiles of the GSE84402 and GSE101685 datasets, including 22 normal and 38 tumor samples, were used as validation cohorts for the prognostic model. The gene expression profiles and clinical data for the ICGC Liver Cancer-RIKEN, JP (LIRI-JP) cohorts, including 232 tumor samples, were also used as validation cohorts for the prognostic model.

A total of 794 anoikis-related genes was acquired from GeneCards (https://www.genecards.org/ (accessed on 22 September 2022)), and 448 of these genes were selected based on a score > 0.4. The overlap of DEGs for LIHC and anoikis-related genes was visualized using the R package “VennDiagram”.

### 4.2. Weighted Gene Coexpression Network Analysis (WGCNA)

WGCNA was performed to identify highly correlated gene modules in TCGA-LIHC [55]. Overlapping genes of DEG for LIHC and anoikis-related genes in TCGA were subjected to WGCNA using the R package “WGCNA”. A total of 422 samples with 168 overlapping differentially expressed anoikis-related genes was used as an expression matrix for further analysis. The soft thresholding power was chosen as 5 to construct a gene network and calculate coexpression similarity and adjacency, which was transformed into a topological overlap matrix (TOM). Hierarchical clustering based on TOM was used to cluster the modules. Finally, the modules that were strongly associated with clinical traits were identified. For intramodular analysis, critical genes were identified in modules with a high gene significance (GS) and module membership (MM), where the correlation between the genes and clinical traits was termed as GS, and the correlation between the module eigengene and the gene expression profiles was termed as MM.

### 4.3. Consensus Clustering Analysis of Anoikis-Related Genes

Unsupervised subgroups and clusters of TCGA-LIHC datasets were identified using the R package “ConsensusClusterPlus” [56], based on overlapping anoikis-related genes and highly correlated module genes from WGCNA. Clusters were verified by Principal Component Analysis (PCA), t-distributed Stochastic Neighbor Embedding (t-SNE) and Uniform Manifold Approximation and Projection (UMAP) using the R packages “broom”, “Rtsne” and “umap”, respectively. The Kaplan–Meier survival curves of the different subgroups and clusters were analyzed and plotted using the R packages “survival” and “survminer”. The gene expression of different subgroups and clusters was visualized using the R package “ComplexHeatmap”.

### 4.4. GSVA of Anoikis-Related Genes

The KEGG pathway and Reactome pathway were analyzed to explore the differences in the biological processes between the different subgroups using the R packages “GSVA” [57] and “msigdbr”. An ssGSEA algorithm was used to investigate the immune cell infiltration relationships between the different subgroups. The infiltration of the immune cells in the different subgroups was visualized using the R package “ggplot2”.

### 4.5. Functional Enrichment Analysis

Biological functional enrichment was analyzed using GO analysis and the KEGG pathway based on The Database for Annotation, Visualization and Integrated Discovery (DAVID) database (https://david.ncifcrf.gov/ (accessed on 29 September 2022)). The cutoff criterion was defined as *p* < 0.05.

### 4.6. Construction of Anoikis-Related Prognostic Risk Model

The gene expression profiles and survival data were merged for further analysis. Finally, a training dataset (n = 183) and a test dataset (n = 182) were randomly sampled from 365 patients at a 1:1 ratio. Univariate Cox proportional hazards regression analysis was performed to screen the candidate prognostic genes and visualize them using the R packages “survival” and “forest plot”. Genes with *p* < 0.05 were included and analyzed with LASSO regression analysis to avoid overfitting using the R package “glmnet” [58]. The risk score was constructed as a prediction factor equal to the summation of coefficients and related genes:$$\mathrm{Risk}\;\mathrm{score}=\sum_{i=1}^{n}Coef_{i}\text{×}X_{i}$$

*Coef*_i_ is the correlation coefficient of the prognostic genes and X_i_ is the expression of the prognostic genes. Finally, multivariate Cox proportional hazards regression analysis was performed to identify the critical clinical phenotypes.

### 4.7. Survival Analysis

According to the median risk score, patients in the training and test datasets were divided into high- and low-risk groups. The prognostic gene expression was plotted as a heatmap using the R package “ComplexHeatmap” for the training and test datasets. The predictive effect of the model was illustrated using Kaplan–Meier survival curves with the R package “survminer”. Different endpoints (1, 3 and 5 years) were set, and the performance of the model was evaluated using time-dependent receiver operating characteristic (ROC) curves using R package “timeROC”. A nomogram was applied to predict the overall survival according to the risk score and clinicopathologic characteristics, such as age, gender and stage, using the R package “rms”. A Sankey diagram was applied to show the cluster distribution of the risk groups and survival outcomes using the R packages “highcharter”, “ggplot2” and “ggalluvial”.

### 4.8. Tumor Immune Analysis

A CIBERSORT algorithm was applied to analyze the correlation between the prognostic genes and the risk score and tumor-infiltrating immune cells.

### 4.9. Validation of Prognostic Risk Model

The prognostic risk model was validated using ICGC-LIRI-JP cohorts, and the gene expression of the prognostic genes was validated using the GSE84402 and GSE101685 datasets. Finally, the protein expression of the prognostic genes in the normal and tumor samples was validated using the Human Protein Atlas (HPA) database (https://www.proteinatlas.org/ (accessed on 3 October 2022)).

### 4.10. Statistical Analysis

All statistical analyses were performed in R software (version 4.2.1) and a *p*-value < 0.05 was considered statistically significant.

## Figures and Tables

**Figure 1 ijms-24-02862-f001:**
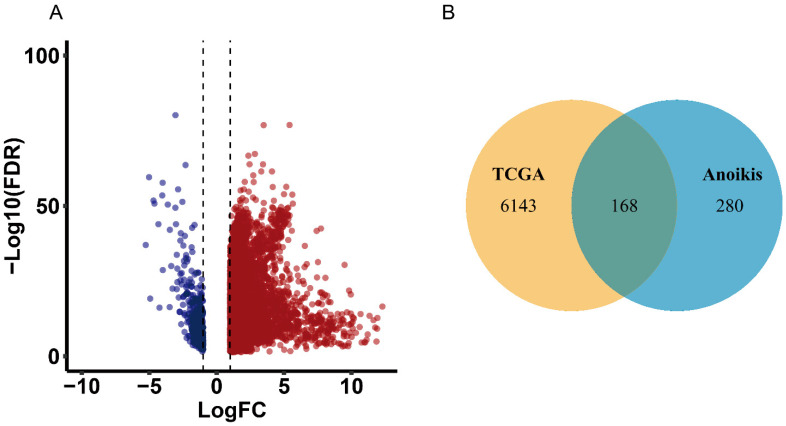
Identification of differentially expressed overlapping anoikis-related genes. (**A**) Volcano plot of differentially expressed genes in the TCGA dataset. Blue dots indicate statistically significant downregulated genes and red dots indicate statistically significant upregulated genes. The *x*-axis represents the logarithm of fold change of differentially expressed genes and the *y*-axis represents the adjusted *p*-value based on the FDR correction method. (**B**) Venn diagram of overlapping genes in TCGA and anoikis-related genes.

**Figure 2 ijms-24-02862-f002:**
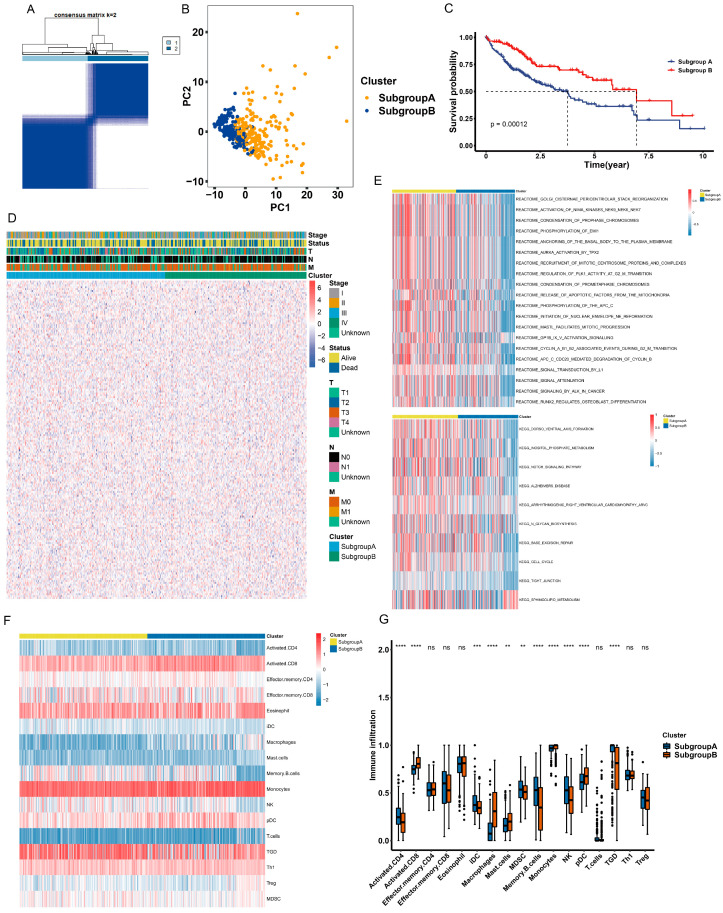
Identification of anoikis-related subgroups. (**A**) Consensus matrix heatmap defining two subgroups. TCGA-LIHC cohorts were divided into two subgroups based on gene expression profiles. The variable k is a parameter used to determine the number of clusters and k = 2 indicates that two subgroups are clustered. Light blue represents “1”, which indicates one of the subgroups, and dark blue represents “2”, which indicates the other subgroup. (**B**) Principal component analysis of the two subgroups. The *x*-axis represents the principal component 1 and the *y*-axis represents the principal component 2. (**C**) The overall survival of the two subgroups. Blue represents patients in subgroup A and red represents patients in subgroup B. The *p*-value is 0.00012. (**D**) Complex heatmap of expression levels of anoikis-related genes in the two subgroups. Blue represents the expression of genes in subgroup A and green represents the expression of genes in subgroup B. Clinicopathologic characteristics, including the tumor stage, status and pathology stage of T, M and N are presented above the complex heatmap. (**E**) Reactome and KEGG pathway analyses of GSVA in the two subgroups. The upper panel shows the enriched Reactome pathway and the lower panel shows the enriched KEGG pathway. Yellow represents subgroup A and blue represents subgroup B. (**F**) The ssGSEA analysis of the immune cell infiltration level in the two subgroups. Yellow represents subgroup A and blue represents subgroup B. (**G**) Boxplot of the abundance of immune cells in the two subgroups. The *x*-axis represents the type of immune cells and the *y*-axis represents the level of immune infiltration. Blue represents subgroup A and orange represents subgroup B. ** *p* < 0.01, *** *p* < 0.001, **** *p* < 0.0001, ns, not significant.

**Figure 3 ijms-24-02862-f003:**
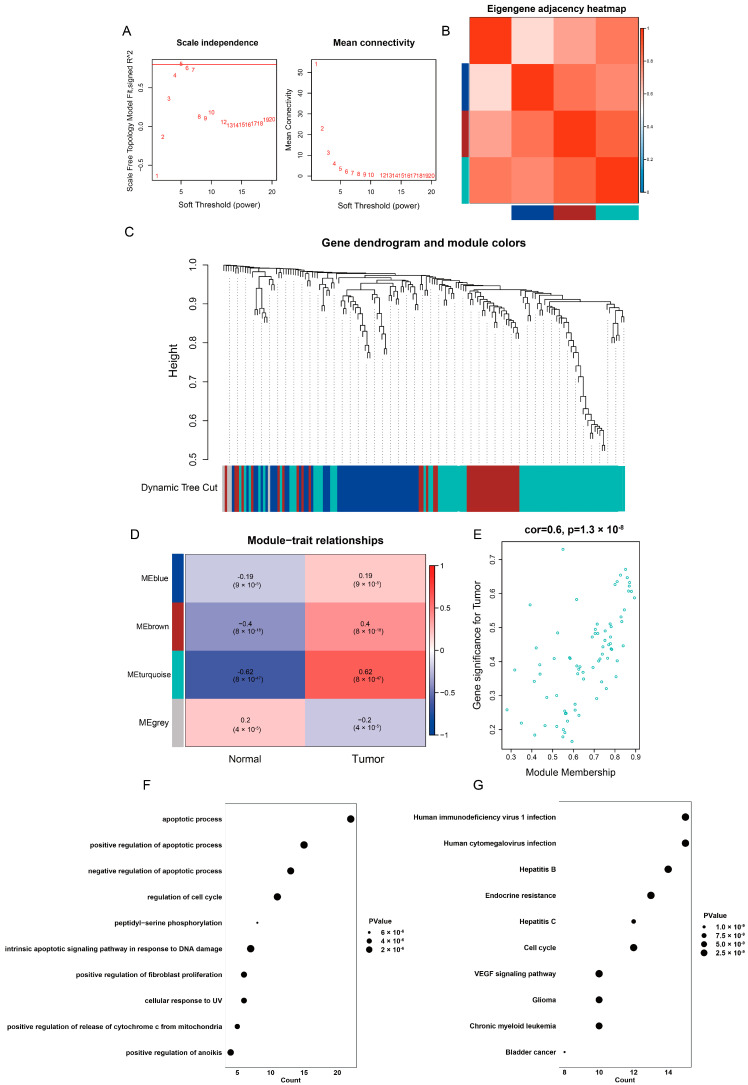
Identification of the highly correlated gene modules in WGCNA. (**A**) Determination of the soft thresholding power. The *y*-axis of the left panel represents the scale-free fit index and the *y*-axis of the right panel represents the mean connectivity. The *x*-axis represents power value. (**B**) Heatmap of Eigengene adjacency. The colors in the heatmap represent the Pearson correlation coefficient between the four gene modules (white, blue, brown and turquoise). The values range from 0 (not correlated) to 1 (highly correlated), marked with blue to dark red. (**C**) Dendrogram of differentially expressed genes clustered based on a dissimilarity measure (1-TOM). The colors represent the identified gene modules. (**D**) Correlations of gene modules with clinical traits. Boolean variables denote the phenotypes of the clinical traits, where 0 represents “Normal” and 1 represents “Tumor”. (**E**) Gene correlation scatter plot of the turquoise module. The *x*-axis represents module membership (MM), which is the correlation between the genes and modules. The *y*-axis represents gene significance (GS), which is the correlation between the genes and clinical traits. The correlation coefficient is 0.6, indicating that the genes significantly associated with tumor are also the central elements of the turquoise modules associated with tumor. The *p*-value is <0.0001. (**F**) GO–Biological Process (BP) term enrichment of genes in the turquoise module. The *x*-axis represents the number of genes in the GO–BP terms and the *y*-axis represents the enriched GO–BP terms. The *p*-value is <0.05. (**G**) KEGG pathway enrichment of genes in the turquoise module. The *x*-axis represents the number of genes in the KEGG pathway and the *y*-axis represents the enriched KEGG pathway. The *p*-value is <0.05.

**Figure 4 ijms-24-02862-f004:**
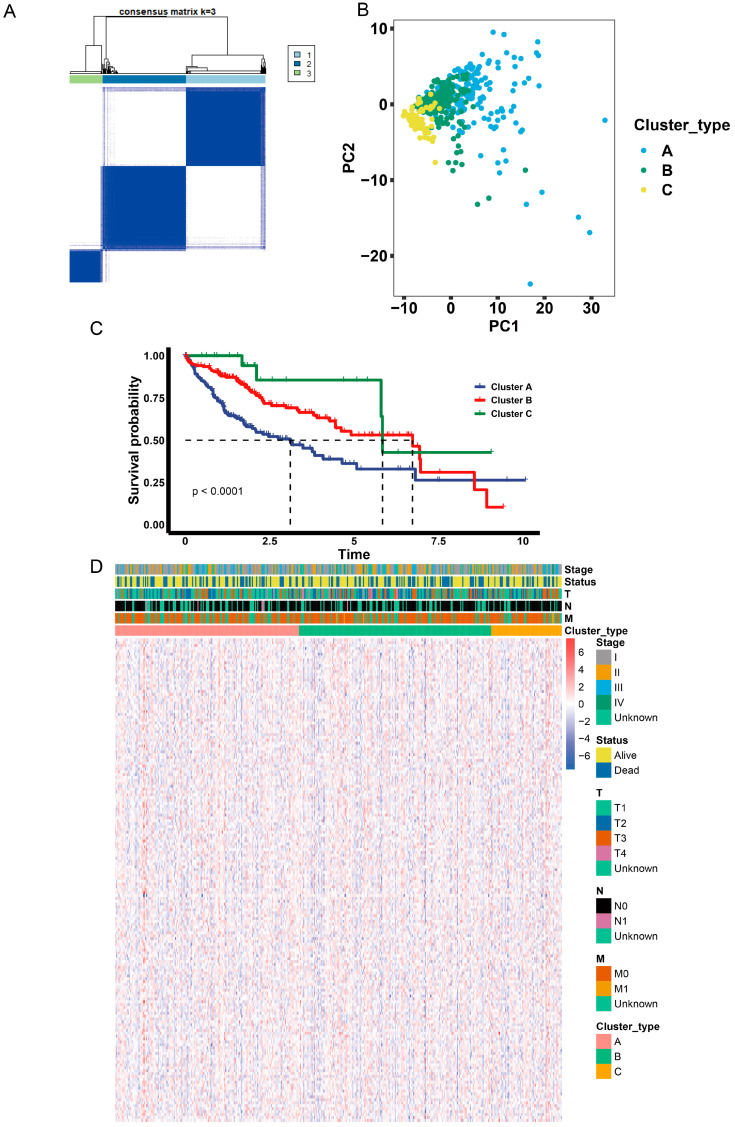
Identification of anoikis-related gene clusters. (**A**) Consensus matrix heatmap defining three gene clusters. The expression k = 3 indicates that three clusters are classified. (**B**). Principal component analysis of the three gene clusters. The *x*-axis represents the principal component 1 and the *y*-axis represents the principal component 2. (**C**) The overall survival of the three gene clusters. Blue represents Cluster A, red represents Cluster B and green represents Cluster C. The *p*-value is <0.0001. (**D**) Complex heatmap of the expression levels of the anoikis-related genes in the three gene clusters. Red represents Cluster A, green represents Cluster B and yellow represents Cluster C. Clinicopathologic characteristics, including the tumor stage, status and pathology stage of T, M and N are presented above the complex heatmap.

**Figure 5 ijms-24-02862-f005:**
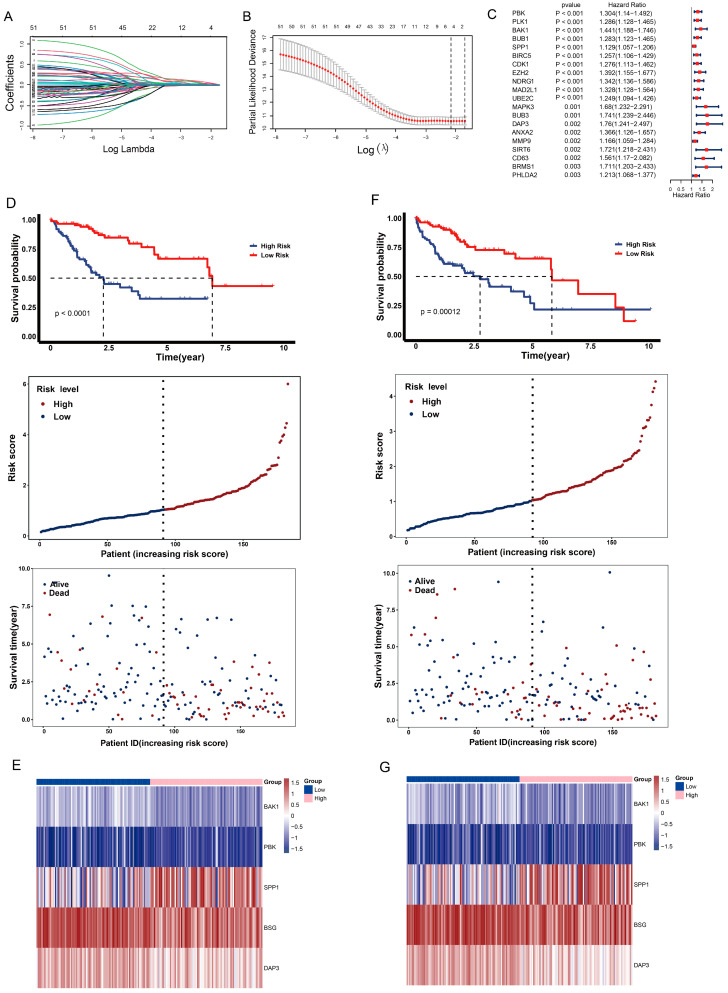
Construction of an anoikis-related prognostic risk model. (**A**) Coefficient curve. Different colors represent different genes. No zero values were selected as a penalty coefficient. (**B**) The minimum lambda of the lasso model was selected via 10 folds of cross-validation. Lambda was determined when the partial likelihood deviance was smallest. (**C**) Forest plot of the top 20 candidate prognostic genes. The 95% confidence interval of the Hazard ratio is displayed. Genes with a hazard ratio > 1 are detrimental prognostic genes. The *p*-value is <0.05. (**D**) The overall survival of the different risk groups, risk score distributions and survival times in the test dataset. Upper panel: Blue represents high-risk patients and red represents low-risk. The patients were divided according to their median risk score. Middle panel: The *x*-axis represents the number of patients in the test dataset and the *y*-axis represents the risk score. Red represents patients in the high level and blue represents patients in the low level. Lower panel: The *x*-axis represents the number of patients and the *y*-axis represents the survival time of the patients in the test dataset. (**E**) Heatmap expression of the prognostic genes in the different risk groups in the test dataset. Blue represents the low-risk group and pink represents the high-risk group. (**F**) The overall survival of the different risk groups, risk score distribution and survival time in the training dataset. Upper panel: Blue represents high-risk patients and red represents low-risk. The patients were divided according to their median risk score. Middle panel: The *x*-axis represents the number of patients in the training dataset and the *y*-axis represents the risk score. Red represents patients in the high level and blue represents patients in the low level. Lower panel: The *x*-axis represents the number of patients and the *y*-axis represents the survival time of the patients in the training dataset. Blue represents living patients and red represents deceased patients. (**G**) Heatmap of the expression of the prognostic genes in the different risk groups in the training dataset. Blue represents the low-risk group and pink represents the high-risk group.

**Figure 6 ijms-24-02862-f006:**
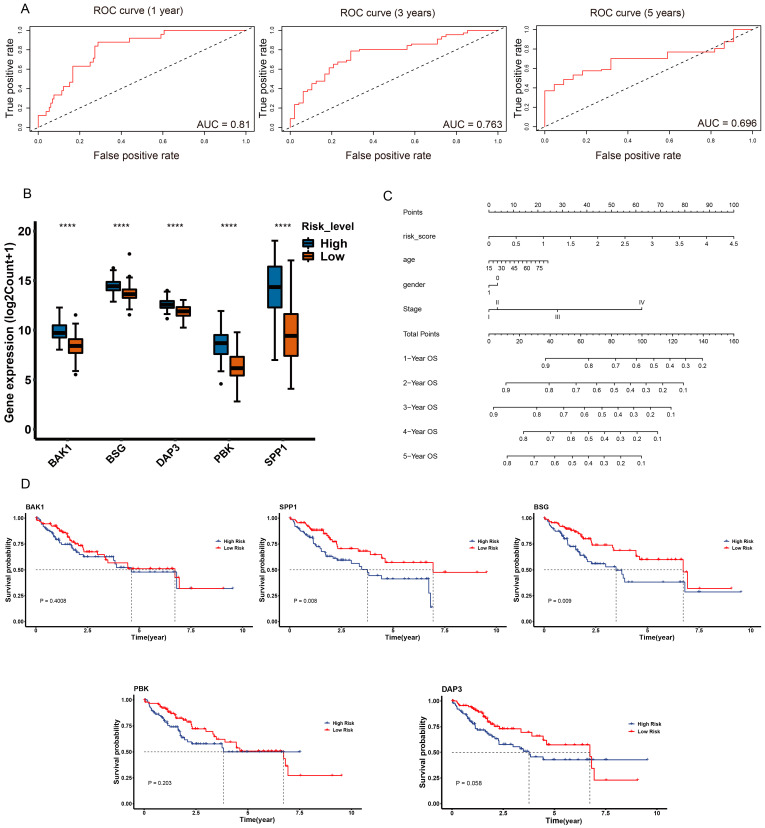
The performance of the prognostic risk model in the test dataset. (**A**) The time-dependent ROC curve of the performance of the prognostic model at 1, 3 and 5 years in the test dataset. The *x*-axis represents the false positive rate and the *y*-axis represents the true positive rate. The AUC value represents the area under the curve. (**B**) Boxplot of the expression of prognostic genes in the different risk groups in the test dataset. The *x*-axis represents the five prognostic genes and the *y*-axis represents the expression level of the prognostic genes. Blue represents the high-risk group and orange represents the low-risk group. (**C**) The nomogram for the prediction of overall survival in LIHC based on the risk score. ”Points” is a scoring scale for individual factors. “Total Points” is the sum of the scoring scale for each factor, such as risk score, age, gender and stage. The overall survival rate of 1–5 years was inferred according to “Total Points”. (**D**). The overall survival of the different risk groups of a single gene, including BAK1, SPP1, BSG, PBK and DAP3 in the test dataset. Blue represents patients in the high-risk group and red represents patients in the low-risk group. **** *p* < 0.0001.

**Figure 7 ijms-24-02862-f007:**
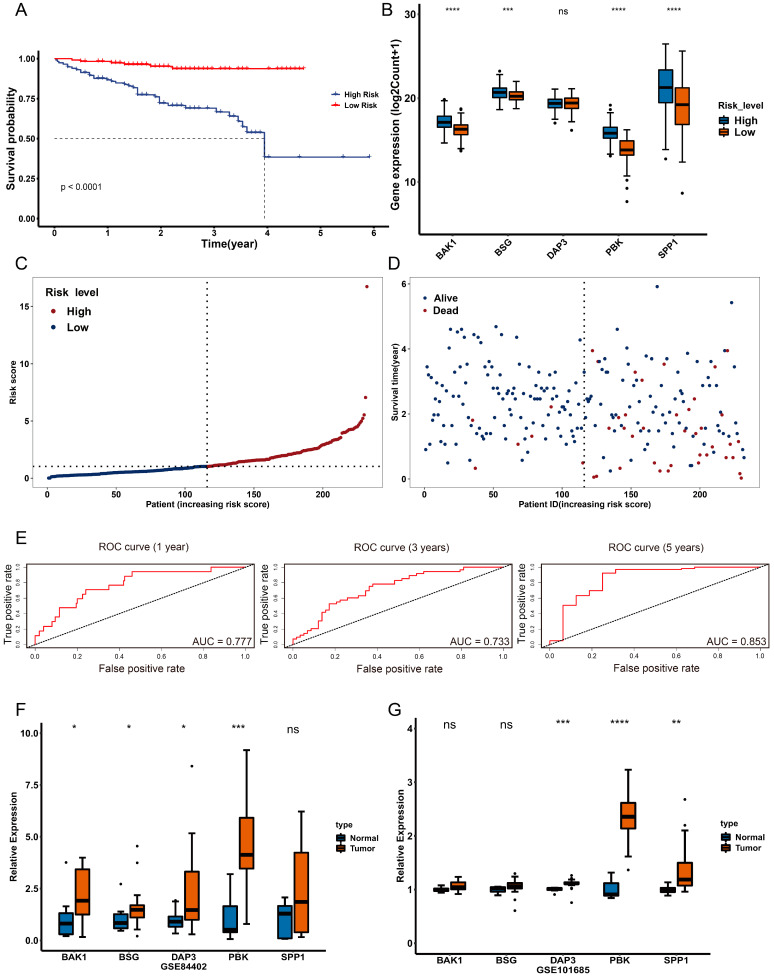
Validation of the anoikis-related prognostic risk model. (**A**) The overall survival of the different risk groups in the ICGC dataset. Blue represents patients in the high-risk group and red represents patients in the low-risk group. (**B**) Boxplot of expression of the prognostic genes in the different risk groups in the ICGC dataset. The *x*-axis represents the five prognostic genes and the *y*-axis represents the expression level of the prognostic genes. Blue represents the high-risk group and orange represents the low-risk group. (**C**) The distribution of the risk score in the ICGC dataset. The *x*-axis represents the number of patients in the ICGC dataset and the *y*-axis represents the risk score. Red represents patients in the high level and blue represents patients in the low level. (**D**) The distribution of the survival time in the ICGC dataset. The *x*-axis represents the number of patients and the *y*-axis represents the survival time of the patients in the ICGC dataset. Blue represents living patients and orange represents deceased patients. (**E**) The time-dependent ROC curve of the performance of the prognostic model at 1, 3 and 5 years in the ICGC dataset. The *x*-axis represents the false positive rate and the *y*-axis represents the true positive rate. The AUC value represents the area under the curve. (**F**,**G**) Boxplot of the expression of prognostic genes in the different risk groups in GSE84402 and GSE101685. The *x*-axis represents the five prognostic genes and the *y*-axis represents the relative expression level of the prognostic genes. * *p* < 0.05, ** *p* < 0.01, *** *p* < 0.001, **** *p* < 0.0001, ns, not significant.

**Figure 8 ijms-24-02862-f008:**
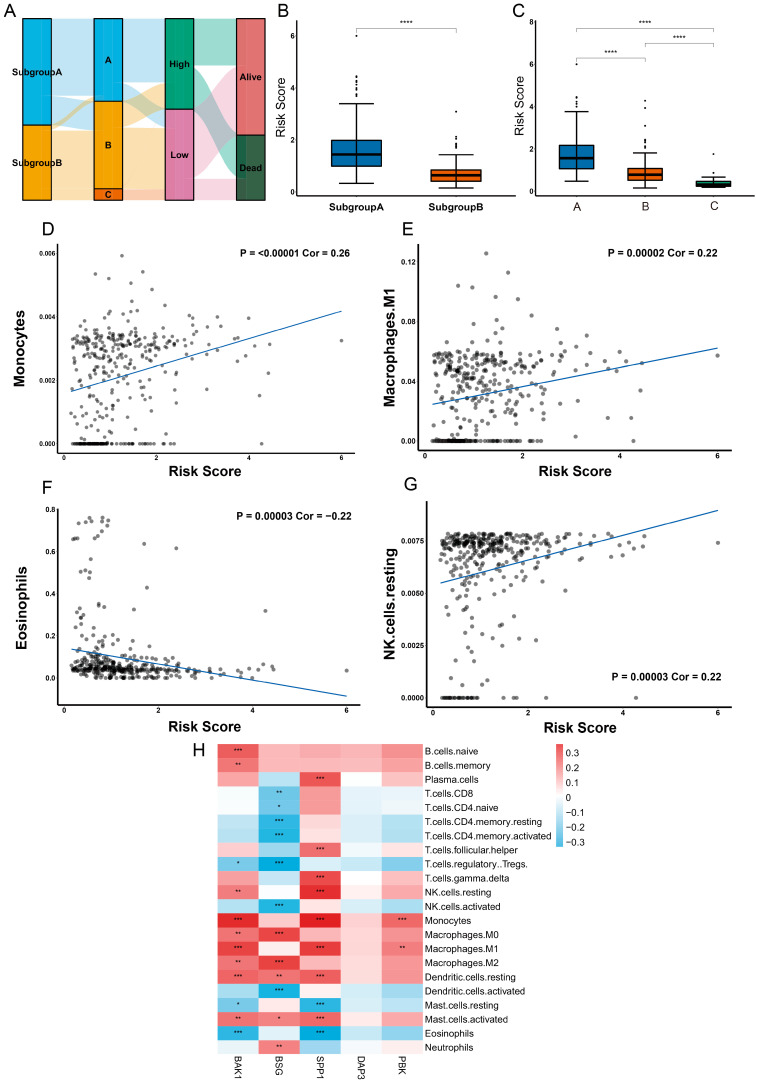
Different characteristics of immune cell infiltration between the two risk groups. (**A**) Sankey diagram of the two subgroups, three gene clusters, two risk groups and two clinical outcomes. Subgroups A and B were subjected to gene clustering, classification according to the two risk levels (high-risk or low-risk) and two clinical outcomes (living or deceased). (**B**) Differences in the risk scores between the two subgroups. The *x*-axis represents the type of subgroup and the *y*-axis represents the risk score. Statistical analysis was performed using Student’s *t*-test. (**C**) Differences in the risk scores between the three gene clusters. The *x*-axis represents the type of subgroup and the *y*-axis represents the risk score. Statistical analysis was performed using one-way analysis of variance. (**D**–**G**) Correlation of the risk score with the immune cells, including (**D**) monocytes, (**E**) macrophage M1, (**F**) eosinophils and (**G**) resting NK cells. (**H**) Correlation of the immune cell infiltration with the prognostic genes. * *p* < 0.05, ** *p* < 0.01, *** *p* < 0.001, **** *p* < 0.0001.

## Data Availability

Publicly available datasets were analyzed in this study. These data can be found here: https://www.cancer.gov/about-nci/organization/ccg/research/structural-genomics/tcga, https://www.ncbi.nlm.nih.gov/gds/ and https://dcc.icgc.org/ (accessed on 23 September 2022).

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
