# Peer review of "Identification of Anoikis-Related Subgroups and Prognosis Model in Liver Hepatocellular Carcinoma"

_ijms, 2023, doi:10.3390/ijms24032862_

Round 1
Reviewer 1 Report
The work entitled "Identification of anoikis-related subgroups and prognosis model in liver hepatocellular carcinoma" done by Chen et.al. focused on finding the mechanism of anoikis, a different form of apoptosis in the progression of liver hepatocellular carcinoma. The authors found a set of 5 genes which are probabilistically associated with the prognosis and were involved in the anoikis process. These set of genes also could correlate with the overall survival of the patients. The author further suggest to elucidate the mechanism by which these gens could act in the HCC.
The authors used this hypothesis based on the fact that there are no available bio markers for the diagnosis of HCC and the one which are currently used are controversial. Firstly, using the bioinformatics approach the authors segregated the anoikis related gene from the TCGA dataset. Further, these were separated using enrichment analysis using GO and KEGG for identifiying the related pathways. Using various regression analysis the authors tried to co-relate this to the survival / risk of the patients which was further supported by infiltrating immune cells.
The authors had a clear hypothesis and the approach used for testing the hypothesis is streamlined by set experimental design. The quality of data presented and the statistical test are done appropriately to the best of my knowledge. The materials and methods section is complete and gives all the necessary information. Finally, the discussion is well written and discusses the possible mechanism, and proposes further studies. However, there are some concerns which are as follows.
As the authors mentioned the role of MAPK in the anoikis the role of ERK1/2 signaling in the HCC resistance is well displayed in the article PMID: 33391517 and should cite it. In the introduction section where the role of TLR is mentioned at the line 58 the author should cite PMID: 29032985 which tells how TLR dictates the tumor fate.
As the authors identified many genes which are involved in the anoikis resistance they should show AEG-1 gene in the risk factor as shown in the article PMID: 24941119 and should discuss it accordingly and cite it.
Although the authors found significantly increased expression of few genes in the two data sets viz. GSE84402 and GSE101685 they are uniformly increased in both the sets how would they explain this discrepancy in the data? Are the selected gene truly having a prognostic value since the expression is different in both the groups.
In the line 41 the authors meant CCND1 or CCDN1? As CCND1 is the one involved in the tumor. The authors should recheck on this.
The authors should correct the line 63 “was applied to treatment metastatic tumors”
In the line 121 “The results indicated that a remarked pattern was between two subgroups...” the authors should mention what was the remarked pattern that was observed. The sentence appears to be incomplete.
The authors should correct the line 136 “was a strong highly correlation of tumor occurrence..”
The author should correct the line 299 “There was a little..”
The author need to correct the line 303 “ could alter a liver environment...”
The author need the complete sentence 327 “the anoikis may related to mitochondrial.”
Line 331 “indicated that the role of BAK was confused in different type of cancers…”
Over all the work done by Chen et.al. is commendable and adds to the necessary information.
Author Response
Point-to-point responses to the Editor’s comments (Manuscript ID: ijms-2062029)
Identification of anoikis-related subgroups and prognosis model in liver hepatocellular carcinoma
Overall comments:
The work entitled "Identification of anoikis-related subgroups and prognosis model in liver hepatocellular carcinoma" done by Chen et.al. focused on finding the mechanism of anoikis, a different form of apoptosis in the progression of liver hepatocellular carcinoma. The authors found a set of 5 genes which are probabilistically associated with the prognosis and were involved in the anoikis process. These set of genes also could correlate with the overall survival of the patients. The author further suggest to elucidate the mechanism by which these gens could act in the HCC.
The authors used this hypothesis based on the fact that there are no available bio markers for the diagnosis of HCC and the one which are currently used are controversial. Firstly, using the bioinformatics approach the authors segregated the anoikis related gene from the TCGA dataset. Further, these were separated using enrichment analysis using GO and KEGG for identifiying the related pathways. Using various regression analysis the authors tried to co-relate this to the survival / risk of the patients which was further supported by infiltrating immune cells.
The authors had a clear hypothesis and the approach used for testing the hypothesis is streamlined by set experimental design. The quality of data presented and the statistical test are done appropriately to the best of my knowledge. The materials and methods section is complete and gives all the necessary information. Finally, the discussion is well written and discusses the possible mechanism, and proposes further studies. However, there are some concerns which are as follows.
- As the authors mentioned the role of MAPK in the anoikis the role of ERK1/2 signaling in the HCC resistance is well displayed in the article PMID: 33391517 and should cite it. In the introduction section where the role of TLR is mentioned at the line 58 the author should cite PMID: 29032985 which tells how TLR dictates the tumor fate.
Response: Thank you for your sincere suggestions. As you suggest, we cited the article (PMID: 33391517) and added it in Introduction in line 57.
In the meantime, we cited the article (PMID: 29032985) and added the role of TLR in Introduction in line 60-61, as follows:
“toll-like receptor (TLR) that recognized damage-associated molecular patterns released from tumorigenic cells could activate T cell and remove tumorigenic cells”
Additionally, Related reference has updated.
- As the authors identified many genes which are involved in the anoikis resistance they should show AEG-1 gene in the risk factor as shown in the article PMID: 24941119 and should discuss it accordingly and cite it.
Response: Thank you for your valuable suggestion. As you suggest, we cited and added it in Discussion in line 264-266, as follows:
“Astrocyte elevated gene-1 (AEG-1) had been identified as an important oncogene that promoted anoikis resistance in HCC cells. AEG-1 promoted anoikis resistance by activating the PI3K/Akt pathway and upregulating the apoptosis protein BCL-2 and the phosphorylation of Bad.”
Additionally, Related reference has updated.
- Although the authors found significantly increased expression of few genes in the two data sets viz. GSE84402 and GSE101685 they are uniformly increased in both the sets how would they explain this discrepancy in the data? Are the selected gene truly having a prognostic value since the expression is different in both the groups.
Response: We really agree with your suggestion. In fact, the expression of five prognostic genes was significantly upregulated in high risk group compared to those in low risk group in TCGA and ICGC datasets, indicating these genes were detrimental genes and may contribute to the development and progression of HCC. They were therefore significantly upregulated in tumor samples compared to normal samples in GSE84402 and GSE101685.
Additionally, there are two reasons why the expression was different in both datasets.
- All the prognostic genes had a growing tendency, although some are statistically significant.
- The quality of two datasets was not uniform which may lead to the difference.
We hope that our response is clear and meet your satisfaction.
Minor concerns:
- In the line 41 the authors meant CCND1 or CCDN1? As CCND1 is the one involved in the tumor. The authors should recheck on this.
Response: Thank you for your suggestion. we are sure that we mean CCDN1
- The authors should correct the line 63 “was applied to treatment metastatic tumors”
Response: Thank you for your comment. We corrected it in the revised manuscript in line 66-67, as follows:
“In the meantime, pharmacological compounds and molecular inhibitors had been used to treat metastatic tumors resulting from anoikis resistance.”
- In the line 121 “The results indicated that a remarked pattern was between two subgroups...” the authors should mention what was the remarked pattern that was observed. The sentence appears to be incomplete.
Response: Thank you for your comment. We corrected it in the revised manuscript in line 115-116, as follows:
“The results indicated that subgroup A had a distinct pattern of immune infiltration compared to subgroup B (Figure 2F).”
- The authors should correct the line 136 “was a strong highly correlation of tumor occurrence..”
Response: Thank you for your suggestion. We corrected it in the revised manuscript in line 129- 130, as follows:
“Among these gene modules, there was a strong correlation between tumor occurrence and the turquoise module”
- The author should correct the line 299 “There was a little..”
Response: Thank you for your comment. We corrected it in the revised manuscript in line 243- 245, as follows:
“However, there was a little research on the relationship between dorsoventral axis formation and liver cancer, suggesting that anoikis-related genes may play a role in tissue and organ development in addition to their involvement in HCC.”
- The author need to correct the line 303 “ could alter a liver environment...”
Response: Thank you for suggestion. We corrected it in the revised manuscript in line 246- 247, as follows:
“Immune cells can alter liver environment and trigger chronic inflammation, eventually leading to the development of hepatocellular carcinoma”
- The author need the complete sentence 327 “the anoikis may related to mitochondrial.”
Response: Thank you for reminder. We added it in the revised manuscript in line 272- 273, as follows:
“This suggested that anoikis may be related to mitochondrial outer membrane permeabilization.”
- Line 331 “indicated that the role of BAK was confused in different type of cancers…”
Response: Thank you for reminder. We corrected it in the revised manuscript in line 276, as follows:
“which indicated that the role of BAK was confused unclear in different types of cancers.”
- Over all the work done by Chen et.al. is commendable and adds to the necessary information.
Response: We sincerely appreciated your approval and we hope that our response is clear and meet your satisfaction.

Reviewer 2 Report
Chen et al. present a method to identify the role of anoikis related genes in HCC and construct a prognostic risk model. While the authors have performed extensive modeling, there are a few points that can addressed to make the article more impactful:
1. The figures need a detailed description of how the plots have been made and the analysis has been done.
2. Have the authors verified any of their claims on any published data on HCC? It would be interesting to perform such an analysis and present in the discussion.
3. In line 48, expand ECM.
4. In line 84, expand DEG and LIHC.
5. In Figure 1B, anoikis is misspelt.
6. In line 97, expand PCA.
7. In line 98, B is in missing after subgroup.
8. In line 109, expand GSVA.
9. In line 137, explain GS and MM.
10. In line 96, explain the meaning of k = 2.
11. In figure 2A, explain the plot and what is 1 and 2?
12. In Figure 3B and 3E, describe the scale bars.
13. Mention the units of gene expression and time in Figure 6.
Author Response
Point-to-point responses to the Editor’s comments (Manuscript ID: ijms-2062029)
Identification of anoikis-related subgroups and prognosis model in liver hepatocellular carcinoma
Chen et al. present a method to identify the role of anoikis related genes in HCC and construct a prognostic risk model. While the authors have performed extensive modeling, there are a few points that can addressed to make the article more impactful:
- The figures need a detailed description of how the plots have been made and the analysis has been done.
Response: we sincerely appreciated your suggestions. In the revised manuscript, we added a large amount of detail description in Figure Legends. We hope that that our supplementary descriptions are clear and meet your satisfaction.
- Have the authors verified any of their claims on any published data on HCC? It would be interesting to perform such an analysis and present in the discussion.
Response: Thank you for your valuable suggestion. As you suggest, we verified our results in published research and discussed it in Discussion, as follows:
BAK in line 277-278
“BAK1 had been identified as a prognostic gene in pyroptosis-related HCC and it associated with persistent hepatitis B virus infection-related HCC”
BSG in line 290-291
“Intriguingly, BSG had been identified as a potential target of non-coding RNA in hepatocellular carcinoma tumorigenesis”
SPP1 in line 292-293
“It had also been identified as a novel biomarker in nonalcoholic steatohepatitis-related HCC”
PBK in line 294-298
“In the progression of HCC, overexpression of PBK had been found to promote HCC cells migration and invasion by activating ETV4/Urokinase-type plasminogen activator receptor signaling pathway [53]. Yang et al. showed that PBK enhanced metastasis of HCC by activating the β-Catenin signaling”
Additionally, related reference has cited and updated.
- In line 48, expand ECM.
Response: Thank you for your comment. We expanded it in line 49
- In line 84, expand DEG and LIHC.
Response: Thank you for your suggestion. We expanded it in line 89.
- In Figure 1B, anoikis is misspelt.
Response: Thank you for your reminder. We corrected it in the revised manuscript.
- In line 97, expand PCA.
Response: Thank you for your comment. We expanded it in line 98-99.
- In line 98, B is in missing after subgroup.
Response: Thank you for your reminder. We added it in the revised manuscript.
- In line 109, expand GSVA.
Response: Thank you for your comment. We expanded it in line 108-109.
- In line 137, explain GS and MM.
Response: Thank you for your valuable suggestion. We clarified them in Results in line 132-134, as follows:
“the gene significance (GS, i.e., the correlation between genes and the clinical traits) and module membership (MM, i.e., the correlation between genes and modules)”
In the meantime, we added it in Figure Legends
- In line 96, explain the meaning of k = 2.
Response: Thank you for your valuable suggestion. k is a parameter to determine the number of clusters, and k = 2 indicates that two subgroups are clustered based on the expression of overlapping anoikis-related genes.
In the meantime. we explained it in Figure Legends in line 623-624, as follows
“k is a parameter to determine the number of clusters, and k = 2 indicates that two subgroups are clustered.”
- In figure 2A, explain the plot and what is 1 and 2?
Response: Thank you for your valuable comment. We classified two subgroups based on the correlation of expression of overlapping anoikis-related genes profiles. Diagonal blue indicates that correlated genes classified TCGA cohorts into two subgroups. Additionally, “1” denoted one of the subgroups and “2” denoted the other subgroup.
In the meantime, we explained it in Figure Legends in line 622-625, as follows:
“TCGA-LIHC cohorts were divided into two subgroups based on gene expression profiles. k is a parameter to determine the number of clusters, and k = 2 indicates that two subgroups are clustered. Light blue represents “1”, which indicate one of the subgroups, and dark blue represents “2”, which indicated the other subgroup.”
- In Figure 3B and 3E, describe the scale bars.
Response: Thank you for your valuable suggestion. we describe it clearly in Figure Legends in line 648-650 and line 655-659, as follows:
“Colors in the heatmap represent the Pearson correlation coefficient between four gene modules (white, blue, brown and turquoise). Values range from 0 (not correlated) to 1 (highly correlated) marked with blue to dark red.”
“X-axis represents module membership (MM), which is the correlation between genes and modules. Y-axis represents gene significance (GS), which is the correlation between genes and clinical traits. The correlation coefficient is 0.6, indicating that genes significantly associated with tumor are also the central elements of turquoise modules associated with tumor.”
- Mention the units of gene expression and time in Figure 6.
Response: Thank you for your reminder. We added the units of gene expression (log2(count+1)) and time (year) in Figure 6.
In the meantime, we revised the units of gene expression and time in all figures.

Round 2
Reviewer 1 Report
The authors have addressed most of the comments raised in the revised manuscript and should check for necessary English language and typos if any.